# Bacterial Strains from Saline Environment Modulate the Expression of Saline Stress-Responsive Genes in Pepper (*Capsicum annuum*)

**DOI:** 10.3390/plants12203576

**Published:** 2023-10-15

**Authors:** María Goretty Caamal-Chan, Abraham Loera-Muro, Reyna De Jesús Romero-Geraldo, Rogelio Ramírez-Serrano

**Affiliations:** 1CONAHCYT-Centro de Investigaciones Biológicas del Noroeste, SC. Instituto Politécnico Nacional 195, Playa Palo de Santa Rita Sur, La Paz C.P. 23096, Baja California Sur, Mexico; mcaamal@cibnor.mx; 2Instituto Tecnológico de La Paz, Boulevard Forjadores de Baja California Sur 4720, 8 de Octubre 2da Secc, La Paz C.P. 23080, Baja California Sur, Mexico; reynaromerog@yahoo.com.mx; 3Centro de Investigaciones Biológicas del Noroeste, SC. Instituto Politécnico Nacional 195, Playa Palo de Santa Rita Sur, La Paz C.P. 23096, Baja California Sur, Mexico; rramirez04@cibnor.mx

**Keywords:** *Capsicum annuum*, salinity stress, plant growth-promoting bacteria, gene expression

## Abstract

Salinity stress is one of the most important problems in crop productivity. Plant growth-promoting bacteria (PGPB) can also confer stress tolerance in plants under saline soil conditions. In a previous work, it was reported that bacteria strains isolated from hypersaline sites mitigated salt stress in chili pepper (*Capsicum annuum* var. Caballero) plants and promoted plant growth in some cases. The aim of this study was to evaluate the modulation of gene expression in *C. annuum* plants by bacteria strains isolated from saline environments. Two bacteria strains from high salinity ponds in Guerrero Negro, BCS, Mexico (*Bacillus* sp. strain 32 and *Staphylococcus* sp. strain 155) and *Azospirillum brasilense* Cd (DSM 1843) were used. Significant improvement in fresh weight yield (stem (28%), root (128.9%), and leaves (20%)) was observed in plants inoculated with *Bacillus* sp. strain 32. qPCR analysis showed that both strains modulated the expression of stress-responsive genes (*MYB*, *ETR1*, *JAR1*, *WRKY*, and *LOX2*) as well as heat shock factors and protein genes (*CahsfA2*, *CahsfA3*, *CahsfB3a*, CaDNaJ02, and CaDNaJ04). Finally, the expression levels of genes related to early salt stress and ISR showed differences in plants with dual treatment (bacteria-inoculated and salt-stressed) compared to plants with simple salinity stress. This work confirmed the differential modification of the transcriptional levels of genes observed in plants inoculated with bacteria under salinity stress.

## 1. Introduction

Salinity stress is one of the major abiotic stresses experienced by plants and one of the most important issues in worldwide crop productivity. The major causes of soil salinity are rising levels of groundwater with high salt content as well as poor-quality drainage and irrigation systems [1]. Stress salinity induces physiological changes in plant cells, such as the generation and accumulation of reactive oxygen species (ROS) functioning as signal molecules; in high concentrations, this has a damaging effect on the plant cells [2]. ROS mechanisms involve antioxidative defense enzymes (i.e., catalase and superoxide dismutase) participating in scavenging and transforming ROS into non-toxic end products, protecting cells from oxidative damage. This antioxidant mechanism is one of the primary responses to stress in plants [3]. On the other hand, abiotic resistance relies on genetic regulation induced by changes in key phytohormone pathways that intersect in a complex manner [4]. This resistance response, or adaptation to salt stress, in the host plant requires the integration and coordination of multiple signals including abscisic acid (ABA), jasmonic acid (JA), gibberellic acid (GA), ethylene, and salicylic acid [1]. Transcriptomic studies have revealed that salt stress causes plant differential expression from hundreds to thousands of genes, some of them categorized as major transcription factor (TF) families, such as MYC2 (master TF of jasmonate signaling), ETR1 (TF that mediates core ethylene signaling), MYB (participate in multiple stress response and ISR), and WRKY (one of the key biological regulators) [1,2,3,4,5,6].

On the other hand, some less characterized proteins such as the DnaJ proteins (heat shock proteins—HSPs) function as molecular chaperones playing critical roles in growth development and multiple stress responses in plants [7]. Such proteins have barely been studied in recent years [7,8]. It is worth mentioning is that heat-shock transcriptional factors (HSFs) regulate HSPs’ expression and can be transcriptionally modulated during salinity stress [9].

Soil salinity is one of the most significant environmental stresses resulting in significant crop reduction. Therefore, it is necessary to improve salinity stress tolerance in crops for global food security [1]. An alternative strategy is the use of microorganisms, such as plant growth-promoting bacteria (PGPB), that have been effective at improving vegetal host, abiotic and biotic, stress tolerance [10]. PGPB can modulate the expression of endogenous genes. Several studies revealed that TF genes are significantly regulated in plants inoculated with PGPB under abiotic stress conditions. These PGPB-activated TF genes are important to regulate downstream stress-response genes, which can alleviate the inhibitory effects of abiotic stress on gene expression [11,12]. Few stress-related genes have been characterized to understand the PGPB-mediated salinity tolerance in the host plant [11]. The bacteria with PGPB-mediated salinity tolerance effects are *Pseudomonas putida* (PS01), which mediates salt tolerance in *Arabidopsis thaliana* and increases defense genes’ expression regulated by the JA pathway via *LOX2* genes [12]; *Dietzia natronolimnaea* that protect *Triticum aestivum* and *Ocimum basilicum* plants under salt stress [9]; and *Bacillus subtilis* (GB03) that enhances salt tolerance in *Arabidopsis thaliana* [12,13].

Physical or chemical changes related to plant defense caused by PGPB are referred to as induced systemic resistance (ISR) that suppress plant diseases caused by a range of pathogens. Also, few reports have already proposed the term “induced systemic tolerance” (IST) for the enhanced tolerance to abiotic stress induced by PGPB [14]. However, PGPB efficiency is determined by different environmental factors like weather conditions, soil characteristics, and interactions with other indigenous bacteria in the soil [14,15]. Bacterial strains isolated from saline habitats are more efficient at enhancing plant salt tolerance than PGPB isolated from non-saline habitats. For instance, the inoculation of *Waha durum* wheat cultivar with *A. brasilense* NH, isolated from saline soil, improved growth under salt-stress conditions [15]. In *Capsicum annuum,* the halotolerant strains *Brevibacterium iodinum* RS16, *Bacillus licheniformis* RS656, and *Zhihengliuela alba* RS111 mitigated salt stress [14]. The search for saline-environment bacteria that can mitigate salt stress in plants is an area of great interest in agriculture, particularly in arid and semi-arid zones.

Peppers (*Capsicum* spp.) are commercially important crops that are cultivated with the use of fertilizers and pesticides, a practice that can increase soil salinity. In addition to pharmaceutical applications, chili peppers (*C. annuum*) can be used as a spice, vegetable, and food coloring. In Mexico, its center of origin, its production was estimated at more than two million tons with a market value of more than USD one billion [16].

In a previous work, it was reported that, from 48 strains isolated from hypersaline sites located in Guerrero Negro, Baja California Sur, Mexico, some strains mitigated salt stress in chili pepper (*C. annuum* var. Caballero) plants, and in some cases, promoted plant growth [17]. In this work, from those 48 strains, 2 were selected for the study of gene expression response of *C. annuum* induced by these strains. Genes involved in enhanced tolerance to abiotic stress (salinity) were targeted.

## 2. Materials and Methods

### 2.1. Bacteria Strains

*Bacillus* sp. strain 32 and *Staphylococcus* sp. strain 155 isolated from hypersaline sites at Exportadora de Sal S.A. (ESSA) in Guerrero Negro, Baja California Sur, Mexico [17], were used along with the *A. brasilense* Cd (DSM 1843) [18].

### 2.2. Evaluation of Bacterial Strains for Growth Promotion in Capsicum annuum

Bacteria strains were tested for plant growth-promoting characteristics. *Capsicum annuum* Var. Caballero was used for the experiment. Seeds were surface-sterilized with 5% hypochlorite solution (*v*/*v*) for 15 min and washed five times with sterile distilled water. The seeds were sown in sterile commercial plant substrate (Sunshine #3, SOGEMIX PM^®^) contained in 50-well trays, cultivated with natural photoperiod and temperature (35 °C day/24 °C night in average) under shade conditions and for 30 days. Plants were irrigated every week with full Hoagland’s nutrient solution and water. Plants that reached approximately 10 cm in height (1 month after seed sowing) were transplanted into individual bags. Four-week-old plants were used for first inoculation.

Bacterial inoculants were obtained from overnight cultures in nutrient broth (NB) at 30 °C and 150 rpm. Five microliters of each bacterial culture were re-inoculated in 200 mL of NB and cultivated at 30 °C and 150 rpm for 72 h. Subsequently, the liquid cultures were adjusted to a bacterial density of 10^6^ CFU/mL with NB. Ten milliliters of the adjusted inoculum were added per plant per treatment. Bacterial inoculation in plants was repeated 15 days after the first application. Plant samples were obtained at the end of the experiment, 30 days after the first inoculation. Treatments corresponding to bacterial inoculation were as follows: *Bacillus* sp. strain 32, *Staphylococcus* sp. strain 155, and *A. brasilense* Cd. Plants were watered to field capacity before bacteria treatments were applied. After, a drip irrigation system was used for the rest of the experiment. Control plants (no bacteria and no saline stress applied) were irrigated with B&D solution [19]. The experiment was performed as a complete randomized block design with six replicates per group. Plants were evaluated at the end of the experiment using the following growth parameters: stem length (SL, cm), stem fresh weight (SFW), stem dry weight (SDW), foliar fresh weight (FFW), foliar dry weight (FDW), root length (RL), root fresh weight (RFW), and root dry weight (RDW). Eight plants per treatment were evaluated at the end of the experiment. Weighing (g) was performed using an analytic balance (Mettler Toledo, AG204), and for dry weights an oven with forced air circulation at 70 °C (Shel-Lab^®^, FX-5, series-1000203) was used until reaching constant weight.

### 2.3. Stress Salinity and Bacteria Strains Induced Expression Genes

#### 2.3.1. Bacterial Inoculation after Salt Stress

Four-week-old plants were used for all treatments. Plants were manually watered to field capacity prior to being exposed to salinity conditions. Salt stress was applied using sterile deionized water containing 50 mM and 100 mM NaCl through a drip irrigation system for two minutes four times a day. Ten mL of the corresponding bacteria treatment were applied after 24 h of salt stress initiation: *Bacillus* sp. strain 32, *Staphylococcus* sp. strain 155, and *A. brasilense* Cd or corresponding mock was root–drench-applied. Plants were organized in a total of 4 blocks. Each block contained the following: (I) plants under salt stress (50 or 100 mM NaCl), (II) a control group without salt stress, (III) plants with the corresponding bacteria strain and no salt stress, and (IV) plants with the corresponding bacteria strain and salt concentrations (32 + 50 or 100 mM NaCl, 155 + 50 or 100 mM NaCl, *A. brasilense* + 50 or 100 mM NaCl). Leaf tissue from a representative group of plants (3 pools with 3 plants each pool), from all treatments, was collected for RNA extraction and differential expression analysis twenty-four hours after bacterial inoculation (T1). Treatments were repeated 15 days after first bacterial inoculation, collecting samples 24 h before (T2) and 24 h after bacteria inoculation (T3). Finally, more samples were collected at the end of the experiment, 30 days after the first inoculation (T4).

#### 2.3.2. Bacterial Strain Inoculation before Salt Stress

The experiment was carried out by first adding the bacterial inoculant and then the salt stress. Prior to bacterial inoculation, plants were manually irrigated to field capacity. Ten mL of the corresponding bacterial treatment was applied 24 h before salt stress. Leaf tissue from a representative group of plants was collected (3 pools with 3 plants each pool), for RNA extraction and differential expression analysis, before salt stress was applied (T1). Afterward, plants were irrigated using sterile deionized water containing 50 mM NaCl solution through a drip irrigation system for two minutes four times a day. Leaf tissue of a representative group of plants from all treatments was collected, for RNA extraction and differential expression analysis, 24 h after the salt stress treatment initiation (48 h after bacterial inoculation, T2).

### 2.4. RNA Extraction

PureZOL RNA isolation reagent (Bio-Rad, Hercules, CA, USA) was used to extract total RNA from leaf tissue samples according to the manufacturer’s specifications. Extracted RNA was quantified with a NanoDrop ND-1000 spectrophotometer (Thermo Fisher Scientific, Waltham, MA, USA). For qRT-PCR analysis, RNA samples were treated with DNase I (1 U μg^−1^ DNA, Thermo Fisher Scientific, Waltham, MA, USA). The absence of DNA was confirmed by performing end-point PCR (40 cycles, according to the real-time PCR program) on the DNase I-treated RNA using Taq-DNA polymerase. Total RNA was stored at −80 °C until qRT-PCR assay.

### 2.5. Relative Gene Expression Levels by qPCR Analysis

Total RNA from leaf tissue samples of *C. annuum* was used for qRT-PCR according to the standard iScriptTM cDNA Synthesis kit (Bio-Rad, Hercules, CA, USA), and the iTaq™ Universal SYBR^®^ Green Supermix (Bio-Rad, Hercules, CA, USA). A “no DNA” template control was used in each analysis. The qPCR conditions were those recommended by the manufacturer: one cycle of pre-treatment at 50 °C for 2 min, one cycle at 95 °C for 10 min, and 40 cycles at 95 °C for 15 s and at 60 °C for 1 min. The reported results are from three independent (*n* = 3) biological replicates. Each biological replicate was tested by triplicate and data were normalized with ubiquitin-conjugating protein (UBI-3) reference gene [20]. Primers used in this work are enlisted in Table 1. The 2^−ΔΔCT^ method was used for relative quantification, where the ΔΔCT value = ((TT_1Target_ − TT_1Reference_) − (CT_1Target_ − CT_1Reference_)). The tested genes were *CaCAT*, *CaSOD(Cu,Zn)*, *CaSOD(Mn)*, *CaNPR1*, *CaMYB72*, *CaETR1*, *CaJAR1*, *CaWRKYa*, *CaLOX2*, *CahsfA2*, *CahsfA3*, *CahsfB3*, *CaDNaJ2*, and *CaDNaJ04*.

### 2.6. Statistical Analysis

Data were analyzed by univariate and multivariate analysis of variance (ANOVA and MANOVA). The least significant differences in Tukey’s HSD test were calculated by two-way ANOVA. Differences among means were considered significant at *p* < 0.05 for all cases. All analyses were conducted using Statistica software version 10.0 for Windows and GraphPad Prism version 6.0 (GraphPad Software, San Diego, CA, USA).

## 3. Results

### 3.1. Effect of Bacteria Isolated from Saline Habitats Inoculation on C. annuum Growth

The bacteria *Bacillus* sp. strain 32, *Staphylococcus* sp. strain 155, and *A. brasilense* were tested for growth promotion effects on the *C. annuum*.

*Bacillus* sp. strain 32 had a positive effect on root dry weight production (63.3%), and for fresh weight a positive effect on the stem (28%), root (128.9%), and leaves (20%) was also observed. All comparisons were performed against the control treatment (*t*-test *p* < 0.05) (Table 2). It is worth noting that *A. brasilense* Cd did not present any effect on plant growth promotion.

### 3.2. Expression Analysis of Stress-Related Genes in Plants Inoculated with the Bacterial Strains from Saline Sites

Transcriptional modification by *Bacillus* sp. strain 32, *Staphylococcus* sp. strain 155, and *A*. *brasilense* Cd in fourteen genes in *C. annuum* was analyzed for their changes in expression (Figure 1 and Figure 2). The effect of bacterial inoculation on the genes’ expression in time 1 showed a null or negative response to all three bacteria in all gene groups analyzed except for the *CaLOX2, Ca MYB72*, *CaDnaJ02*, *CaDnaJ04*, *CaWRKY*, and *CaHsfA2* genes. For the *CaHsfA2* gene, this upregulation proved to be significant at time 1 (Figure 2d). A positive effect in the expression of all genes was observed in time 2 by *Bacillus* sp. strain 32 and *Staphylococcus* sp. strain 155 (Figure 1 and Figure 2). The increase was significant in the *CaJAR1* gene with *Bacillus* sp. strain 32 and the CaDnaJ04 gene with *Staphylococcus* sp. strain 155. The exception was for the *CaLOX2* gene which showed downregulation when plants were inoculated with *Bacillus* sp. strain 32 (Figure 1f). For plants inoculated with *A. brasilense* Cd, all showed null or positive changes in the level of expression, being lower, but not significantly, than those observed with the other inoculants. At time 3, greater effects were observed due bacterial inoculants. Overall, the three bacteria induced positive effects, being significant for *CaCAT*, *CaLOX2*, *CaNPR1*, *CaDnaJ02*, *CaHsfA3,* and *CaHsfB3a* genes with *Bacillus* sp. strain 32, and *CaCAT*, *CaSOD* (*CuZn*), *CaLOX2,* and *CaNPR1* with *A. brasilense* (Figure 1 and Figure 2). However, it was observed that the three inoculated bacteria caused a downregulation of the *CaMYB72* gene and *Bacillus* sp. strain 32 also caused a downregulation of the *CaWRKY* and *CaHsfA2* genes (Figure 1g,h and Figure 2d). Finally, during time 4, the three inoculants caused a null expression or downregulation in most of the genes, becoming significant in *CaCAT*, *CaSOD(CuZn)*, *CaJAR1*, *CaETR1*, *CaLOX2*, *CaMYB72*, *CaWRKY*, *CaNPR1*, *CaHsfA2*, and *CaHsfB3a*.

### 3.3. Effect of Different NaCl Concentrations on Gene Expression of C. annuum

To determine the effects of salt stress in *C. annuum*, plants were exposed to 50 and 100 mM of NaCl treatment. Plant tissue was collected after 48 h for analysis (Figure 3 and Figure 4 to 50 mM, and S3 and S4 to 100 mM of NaCl). A greater response in terms of relative expression was observed under 50 mM of saline stress than at 100 mM. In general, an upregulation was observed in the oxidative stress genes group, being higher at times 3 and 4 (Figure 3a–c). For the second group of genes involved in response to both abiotic and biotic stress, the *CaJAR1* and *ETR1* genes showed downregulation in time 1, to subsequently increase their expression in later times, especially in time 3. For the *CaLOX2* gene, an upregulation was observed in times 1 and 3, and downregulation in times 2 and 4 (Figure 3d–f). In the next group, which includes transcriptional regulators of ISR, an apparent up and downregulation of these genes was observed at different times. An upregulation of the *CaMYB72* gene was observed at time 1 and for *CaWRKY* at time 2 and 3. On the other hand, a downregulation at time 3 was observed for *CaMYB72*, and for *CaWRKY* at times 2 and 4 (Figure 3g,h). Finally, for the *CaNPR1* gene and the groups of HSP and HSF genes (Figure 4), an upregulation was observed, being significantly higher at time 3 for the group of HSP and HSF (Figure 4d). These results indicate that treatment with 50 mM NaCl induces a response in plants detectable up to 48 h after treatment (Figure 3 and Figure 4).

### 3.4. Expression Analysis of Genes Associated with Salt Tolerance in Plants with Dual Treatment

#### 3.4.1. Bacterial Inoculation after Salt Stress

Plants were inoculated with the corresponding bacteria 24 h after being treated with 50 mM NaCl (Figure 5 and Figure 6). In general, a similar response was observed for dual treatment samples (stress + bacteria) in comparison with those treated only with saline stress. The first analyzed group was that of the genes related to the oxidative stress response (*CaCAT*, *CaSOD(CuZn)*, and *CaSOD(Mn)*). Generally speaking, in T1 no changes in expression were observed, except in certain cases wherein there was an increase in expression (Figure 5a–c). For T2, increases in the expression of these three genes were observed, being greater in dual treatments. For T3, the highest increase in the expression levels of the *CaCAT*, *CaSOD(CuZn)*, and *CaSOD(Mn)* genes was observed. Finally, in T4 the expression levels of these genes were similar to those observed in T2, except for *CaCAT* treated with saline stress + strain 155 wherein the greatest increase in expression levels was observed.

The genes involved in response to both abiotic and biotic stress (*CaJAR1*, *CaLOX2*, and *CaETR1*) showed a similar behavior between the four applied treatments. However, a downregulation of the three genes’ expression by bacterial inoculants was observed at T3. The genes related to transcriptional regulators of ISR response (*CaMYB72* and *CaWRKY*) showed some changes between treatments. The *CaMYB72* increased its expression with the inoculants and when subjected to saline stress. The greatest increase in expression was with *A. brasilense* Cd (Figure 5g). In the case of *CaWRKY*, similar behavior was observed in its expression with all treatments (Figure 5h). The transcriptional coregulators *NPR1*, HSP, and HSF showed similar expression levels in all four treatments (Figure 6). Finally, *CaDnaJ02*, *CahsfA2,* and *CahsfB3a* showed a pattern of downregulation due to the effect of bacterial inoculums on T3 similar to that observed in *CaJAR1*, *CaLOX2,* and *CaETR1* (Figure 6b,d,f).

#### 3.4.2. Bacterial Strain Inoculation before Salt Stress

The transcriptional expression of genes related to early salt stress responses such as *CaMYB* (transcription factor), *CaJAR1* (JA signaling), and *CaLOX2* (JA synthesis) was remarkably upregulated in salt stress compared to the control; similar expression patterns were observed in the genes encoding for HSF *CahsfA2* and *CahsfA3*. The analyses show no differences in *CaETR1*, *CaDNaJ2*, and *CaDNaJ4* expression in seedlings under NaCl treatment. By contrast, in dual treatment, *CaJAR1*, *CaMYB*, *CaETR1* and genes encoding for heat shock proteins were downregulated. It is worth noting that the expression levels of *CaLOX2* in dual-treatment plants with *B. pumilus* and *A. brasilense* Cd were higher than with saline stress only (Appendix A).

## 4. Discussion

The use of bacteria with the capacity to promote plant growth (PGPB) is a potential biological tool to alleviate abiotic stress, including salinity. Under highly saline conditions, bacterial diversity is reduced in comparison with other environments [25,26,27]. Nevertheless, bacteria isolated from hypersaline sites are candidates for plant growth promotion and more efficient at enhancing plant tolerance to salt stress than those isolated from non-saline habitats [28]. The effects of PGPB with the rapid activation of transcriptional expression of genes related to stress response led to a faster and stronger primed stress response at the transcriptional level.

In this study, two bacteria strains from ponds with high salinity from Guerrero Negro, Baja California Sur, Mexico, were used (*Bacillus* sp. strain 32 and *Staphylococcus* sp. strain 155). *Bacillus* is one of the most abundant and diverse bacteria in agroecosystems including extreme hypersaline environments. Moreover, *Staphylococcus* was isolated from marine environments in the East coast of India, salinity soils of Sharkia (Egypt), halophyte plants, or fermented fish [29,30,31,32,33,34,35].

This present study demonstrated that *Bacillus* sp. strain 32 can stimulate the growth of *C. annuum* under shade conditions, which is not the case for *Staphylococcus* sp. strain 155 or PGPB *A. brasilense* Cd (Table 2). *Bacillus* sp. has been reported to promote growth in some plant species, and *C. annuum* is among them [36]. Additionally, *B. pumilus* has been reported to promote plant growth in *Solanum lycopersicum*, *Rose hibrida*, *Vicia faba*, *Oryza sativa*, and *Atriplex lentiformis* [25,37,38,39,40]. The above indicates that *Bacillus* has a positive effect on the growth of a wide range of plants despite being isolated from extreme environments, as seen in this work. On the other hand, very little is known about the *Staphylococcus* effect on plant growth. Few reports registered a positive effect on the root lengths of *A. thaliana* (*Brassicaceae* family) [35]. In another study, it was observed that *S. equorum* did not significantly affect growth promotion in *S. lycopersicum* (*Solanaceae* family) [41], which is similar to results obtained in the present work with *Staphylococcus* strain 155 in *C. annuum*. As other PGPB, the growth promotion effect by the *Staphylococcus* genus could be specific to plant species.

Multiple genes have been reported to be involved in plant responses to salinity stress where the expression level depends on NaCl concentration and stress time exposure [42,43]. In this study, salinity response and ISR genes (*CaSOD(Cu,Zn)*, *CaSOD(Mn)*, *CaCAT*, *CaMYB*, *CaWRKY*, *CaJAR1*, and *CaLOX2*) were stimulated by NaCl 50 mM 48 h after saline stress was initiated; these results are consistent with previous reports [5,11,44,45].

It is hypothesized that stress tolerance driven by PGPB is accompanied by bacteria-primed transcriptional activation of multiple stress-responsive factors before plant exposure to salt stresses, in a similar response to ISR against pathogens [46]. In this work was observed a differential modulation in the genes’ expression related to salt-stress responses. The PGPB *A. brasilense* Cd upregulated the expression of *CaMYB*, *CaJAR1*, *CaWRKY,* and *CaLOX2* genes, while *Bacillus* sp. strain 32 and *Staphylococcus* sp. strain 155 downregulated them or did not affect them 24 h after inoculation. *A. brasilense* Cd did not act as a PGPB, at least under current experimental conditions, but activated genes related to the ISR-type response (*CaMYB72*). Nevertheless, molecular mechanisms and the main signaling pathways involved in the defense response triggered by beneficial microorganisms have not been well characterized and vary from one beneficial bacterial species to another [47]. An example is the transcription factor *MYB72* that is suggested as a node of convergence in the ISR signaling pathway triggered by different beneficial microbes as *Pseudomonas* spp., *Trichoderma* spp., and halotolerant *Dietzia natronolimnaea* [11,46,47]. In this study, plants inoculated with bacteria and treated with saline stress did not modify the expression of its genes, except in time 3, where a downregulation was observed. The results obtained in the present study are similar to those obtained by other authors in response to similar treatments (bacteria + stress), where bacteria conferred tolerance to stress [12,48,49,50].

HSFs regulate the expression of heat shock proteins (HSPs) and other chaperone genes. In *Arabidopsis*, the *hsfA4A* transcriptional factor binds to promoters of target genes encoding the small heat shock protein HSP17.6A, as well as the *WRKY30* and *ZAT12* genes, indicating that *hsfA4A*’s role is as a stress-response downstream transcription regulator [10,51]. The ROS controlled by the *HSFA4A* gene are induced by salinity, elevated temperature, and a combination of these conditions [51]. Little is known about the Hsf family in *Capsicum*. Some Hsf were upregulated under salt stress [21]. *CaHsfA3* presented a similar pattern of expression in this work with NaCl 50 mM after 24 h. It is worth noting that, in this study, the *CahsfA2* gene expression level was significantly higher in plants under salinity stress, and this gene has been previously reported to be downregulated under salinity conditions (NaCl) [21].

Evidence of the role of DnaJ proteins (heat shock proteins) on stress response in plants has increased in recent years. Cis-elements in the promoter regions of *CaDnaJ* genes have been identified. The major groups of cis-elements include stress response and hormone response (SA, IAA, GA, MeJA, ABA, and ethylene) [7]. In the present work, only *A*. *brasilense* induced the expression of *CaDnaJ02* and *CaDnaJ04* genes under saline-stress conditions. CaDnaJ02 and CaDnaJ04 genes have cis-response elements to the MeJa hormone in their promoter region [7]. The expression of the genes related to this phytohormone was observed when inoculating them with the PGPB *A. brasilense*. JA synthesis pathways are important during the interaction with beneficial bacteria and salinity stress [33].

## 5. Conclusions

In the present study, we evaluated the gene expression in *C. annuum* plants induced by *Bacillus* sp. strain 32, *Staphylococcus* sp. strain 155 (both isolated from saline environment), and PGPB *A. brasilense*, through RT-qPCR. This study determined that the transcription of *CaJAR1*, *CaMYB72*, WRKY, and *CaLOX2* genes was induced by strains; meanwhile, only the *CaLOX2* gene was involved in the response to dual treatments with all three strains. Such results indicate that the JA signaling pathway is involved in the response to treatments. However, further studies are needed to confirm that the JA pathway is responsible for the effect observed in the strains. These results contribute to improve our understanding of the molecular mechanisms involved when plants are inoculated with bacteria.

## Figures and Tables

**Figure 1 plants-12-03576-f001:**
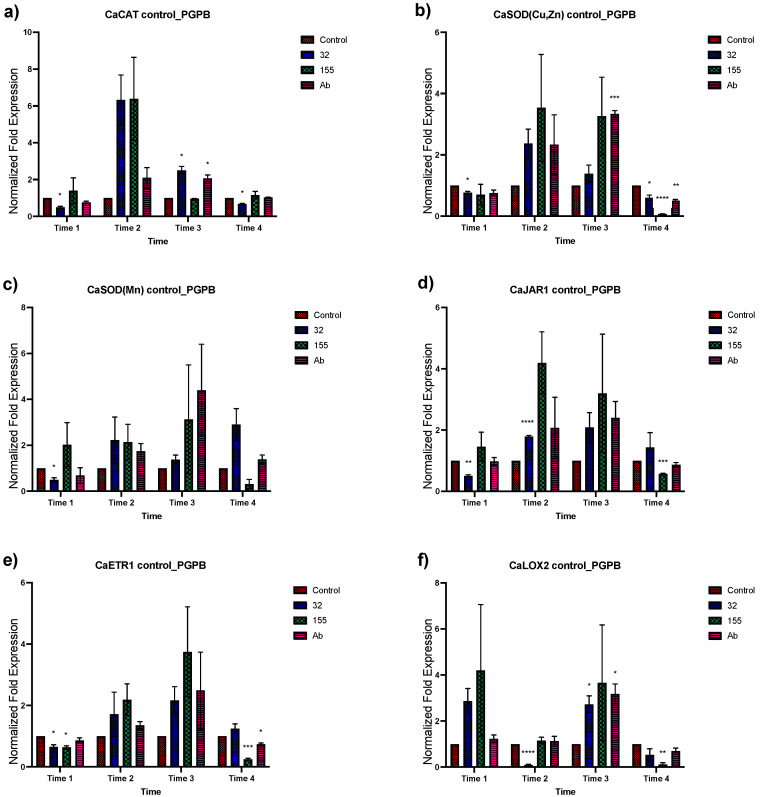
Effects of bacterial inoculation on *C. annuum* plants in genes involved in stress response. Quantitative RT-PCR determinations of genes relative expression levels: *CaCAT*, *CaSOD(Cu,Zn)*, *CaSOD(Mn)*, *CaJAR1*, *CaETR1*, *CaLOX2*, *CaMYB72*, and *CaWRKYa*. The data represent means of biological triplicates and experimental replicates; error bars represent SEM. The asterisk indicates statistically significant differences between treatments (range test *p* < 0.05 *, 0.025 **, 0.01 ***, and 0.001 ****).

**Figure 2 plants-12-03576-f002:**
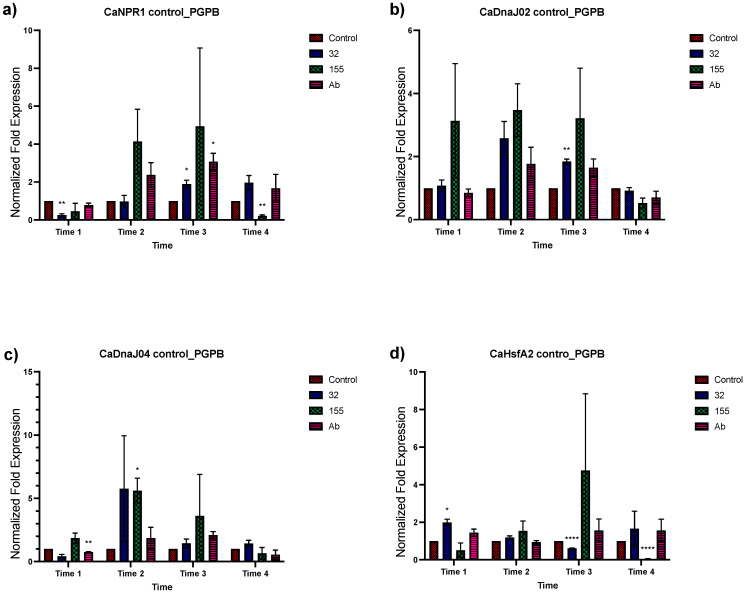
Effects of bacterial inoculation on *C. annuum* plants in NPR1, heat shock protein, and factor genes that regulate genes that encode heat shock proteins. Quantitative RT-PCR determinations of genes relative expression levels: *CaNPR1*, *CaDNaJ2*, *CaDNaJ04*, *CahsfA2*, *CahsfA3*, and *CahsfB3*. Data represent means of biological triplicates and experimental replicates; error bars represent SEM. The asterisk indicates statistically significant differences between treatments (range test *p* < 0.05 *, 0.025 **, and 0.001 ****).

**Figure 3 plants-12-03576-f003:**
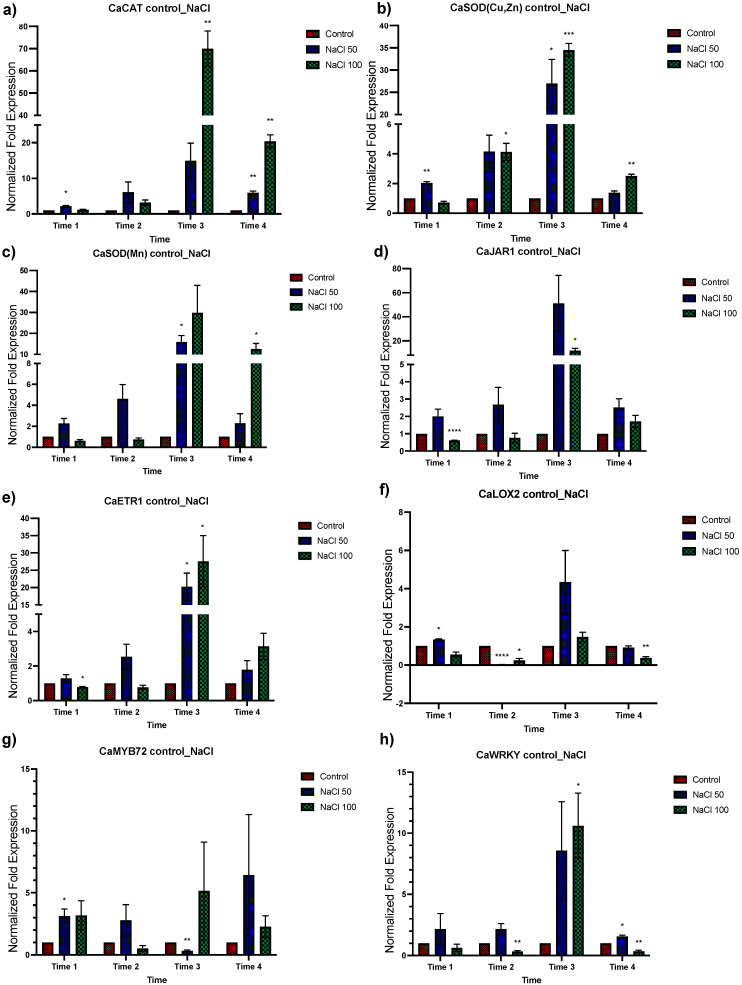
Effects of salt concentration in *C. annuum* plants in genes involved in stress response. Quantitative RT-PCR determinations of genes’ relative expression levels: *CaCAT*, *CaSOD(Cu,Zn)*, *CaSOD(Mn)*, *CaJAR1*, *CaETR1*, *CaLOX2*, *CaMYB72*, and *CaWRKYa*. The data represent means of biological triplicates and experimental replicates; error bars represent SEM. The asterisk indicates statistically significant differences between treatments (range test *p* < 0.05 *, 0.025 **, 0.01 ***, and 0.001 ****).

**Figure 4 plants-12-03576-f004:**
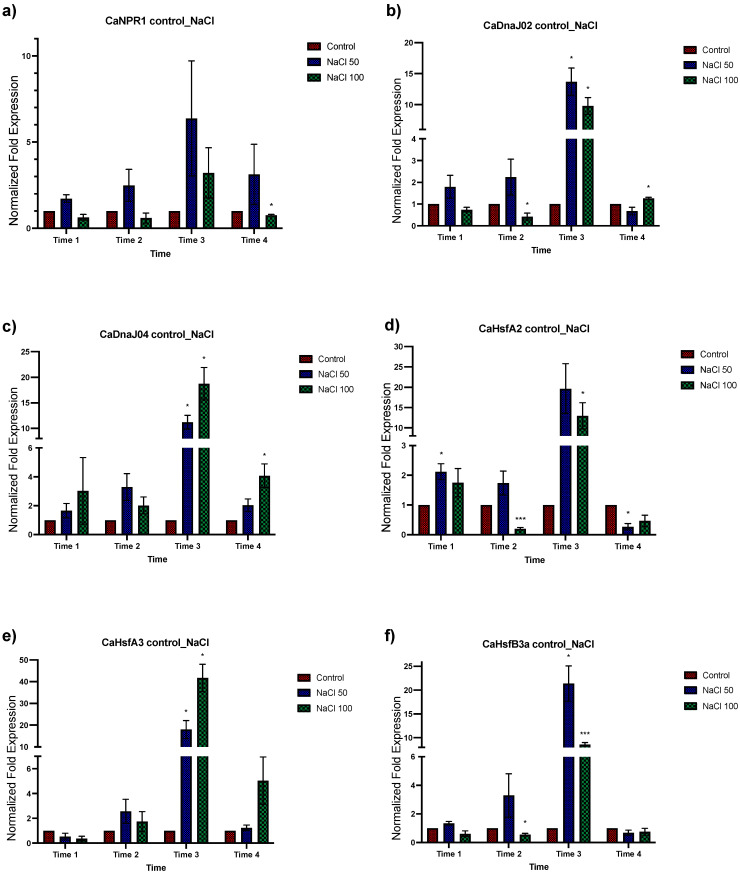
Effects of salt concentration on *C. annuum* plants in NPR1, heat shock protein, and factor genes that regulate genes that encode heat shock proteins. Quantitative RT-PCR determinations of relative expression levels of the genes: *CaNPR1*, *CaDNaJ2*, *CaDNaJ04*, *CahsfA2*, *CahsfA3*, and *CahsfB3*. The data represent means of biological triplicates and experimental replicates; error bars represent SEM. The asterisk indicates statistically significant differences between treatments (range test *p* < 0.05 *, and 0.01 ***).

**Figure 5 plants-12-03576-f005:**
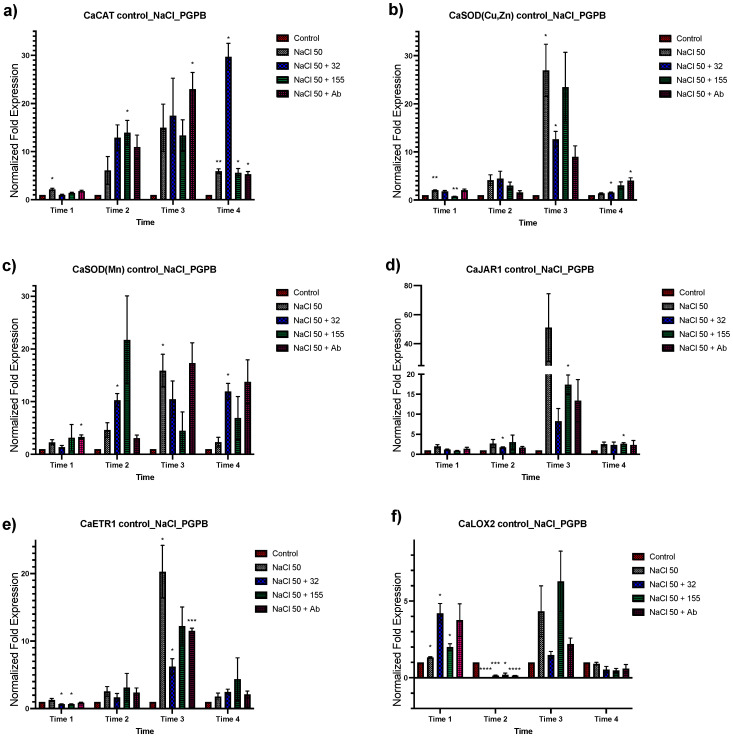
Expression analysis of stress-response genes of *C. annuum* plants inoculated with bacteria after 24 h of salt stress (NaCl 50 mM). Quantitative RT-PCR determinations of genes relative expression levels: *CaCAT*, *CaSOD(Cu,Zn)*, *CaSOD(Mn)*, *CaJAR1*, *CaETR1*, *CaLOX2, CaMYB72*, and *CaWRKYa*. The data represent means of biological triplicates and experimental replicates; error bars represent SEM. The asterisk indicates statistically significant differences between treatments (range test *p* < 0.05 *, 0.025 **, 0.01 ***, and 0.001 ****).

**Figure 6 plants-12-03576-f006:**
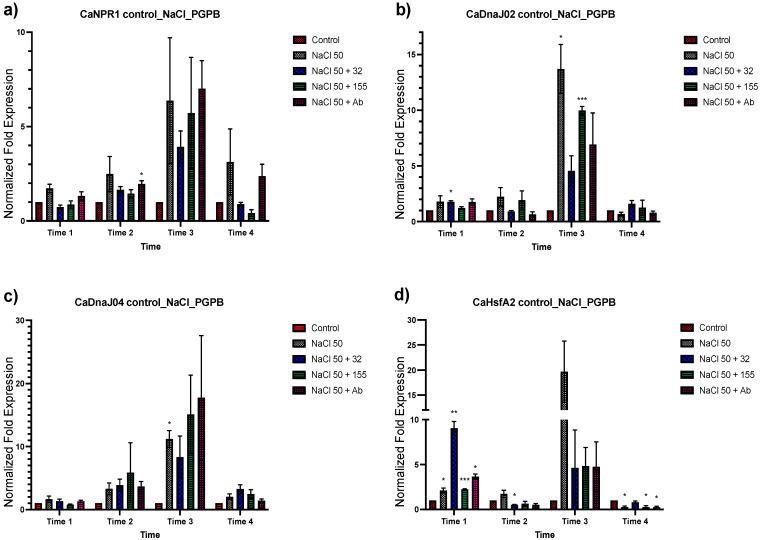
Gene expression analysis of NPR1, heat shock protein genes, and factor genes that regulate genes that encode heat shock proteins of *C. annuum* plants inoculated with bacteria after 24 h of salt stress (50 mM NaCl). Quantitative RT-PCR determinations of relative expression levels of the genes: *CaNPR1*, *CaDNaJ2*, *CaDNaJ04*, *CahsfA2*, *CahsfA3*, and *CahsfB3*. The data represent means of biological triplicates and experimental replicates; error bars represent SEM. The asterisk indicates statistically significant differences between treatments (range test *p* < 0.05 *, 0.025 **, 0.01 ***, and 0.001 ****).

**Table 1 plants-12-03576-t001:** Primers used in this work.

Name	Primer Sequence	References
*CaDnaJ02-F* *CaDnaJ02-R*	5′-GCTGGTCATGCATCTGCTGT-3′ 5′-CACTTGGCCGTTTGGACGAA-3′	[7]
*CaDnaJ04-F* *CaDnaJ04-R*	5′-GGCTTGTGTGCCAAAGGCTA-3′ 5′-GGCCGCAGCTACCAGACTTA-3′	[7]
*CaHsfA2-F* *CaHsfA2-R*	5′-GTAGCATCAGTAGCCACAGC-3′ 5′-CAAGCAACTCTTCCCAAATA-3′	[21]
*CaHsfA3-F* *CaHsfA3-R*	5′-CGAAAGTATGATGAAAGAAGAGG-3′ 5′-ATAGTTGCCAAGACCACCC-3′	[21]
*CaHsfB3a-F* *CaHsfB3a-R*	5′-CGACCGACGACATCGTTT C-3′ 5′-TTGTCATTGCTGAACTCCC-3′	[21]
*CaUBI-3-F* *CaUBI-3-R*	5′-TGTCCATCTGCTCTCTGTTG-3′ 5′-CACCCCAAGCACAATAAGAC-3′	[20]
*CaMYB72-F* *CaMYB72-R*	5′-GTCCTCTGGAGTGAGGAAAGGTGCATGGACTGA-3′ 5′-ATCTATTCAGACCAGCTCTAATGGGAACAAGATGCCACTTTCCT-3′	Designed in this work
*CaWRKYa-F* *CaWRKYa-R*	5′-AATTACGAATTCAATTAACAAAGAT-3′ 5′-ATGGAAGAGTATTGGAATTGTTA-3′	[22]
*CaLox2-F* *CaLox2-R*	5′-CGAGCTGTAGTTACGGTAAGGAACAAGAACAAGGAAGATCTG-3′ 5′-GTGTTTGGATCGATGTCGGTGCTGATGAGTTCTAAGGCG-3′	Designed in this work
*CaETR1-F* *CaETR1-R*	5′-CCACATCATTCCTGATTTACTTAGCGTCAAAACTAGGGAG-3′ 5′-CATTCTAACATGTCTACCTGTCTCCTCTTGAGTCCGAATAATACCCA-3′	Designed in this work
*CaJAR1-F* *CaJAR1-R*	5′-CCCTCAGACTTTTAAGGCTTGTGTTCCTCTTGTCACTCA-3′ 5′-CTTCCCTGAGTGGTACCAGAACTTAATGAGATGGTTGTAATG-3′	Designed in this work
*CaNPR1-F* *CaNPR1-R*	5′-GCACAGAGGACAACAGTGGA-3′ 5′-TCAGTGAACGCTTTGGTCAG-3′	[23]
*CaCAT-F* *CaCAT-R*	5′-GTCCATGAGCGTGGAAGCCCCGAAT-3′ 5′-CGCGATGCATGAAGTTCATGGCACC-3′	[24]
*CaSOD(Mn)-F* *CaSOD(Mn)-R*	5′-CTCTGCCATAGACACCAACTT-3′ 5′-CCAAGTTCGGTCCTTTAATAA-3′	[21]
*CaSOD(Cu,Zn)-F* *CaSOD(Cu,Zn)-R*	5′-GTCCTTAGCAGCAGTGAATGTGTTAGTGGCACCATCCTC-3′ 5′-GCCATGAAGTCCAGGTTTTAGGCCAGAGACATTTCCGGTAACTG-3′	Designed in this work

**Table 2 plants-12-03576-t002:** Stem, root, and leaf length as well as fresh and dry weight of *C. annuum* plants inoculated with potentially growth-promoting isolates (strain 32, 155) and PGPB *A. brasilense*.

Treat	Length	(cm)	Fresh	Weight	(g Plant^−1^)	Dry	Weight	(g Plant^−1^)
	Stem	Root	Stem	Root	Leaf	Stem	Root	Leaf
Not bacteria	24.30 ± 1.125	24.40 ± 0.67	2.5 ± 0.22	3.8 ± 0.33	4.5 ± 0.31	0.5 ± 0.0	0.6 ± 0.1	2.3 ± 0.09
Strain AB	25.80 ± 0.7348	19.6 ± 0.92 **	2.8 ± 0.12	5.4 ± 0.96	4.7 ± 0.2	0.5 ± 0.0	0.7 ± 0.12	2.5 ± 0.09
Strain 32	28.50 ± 0.73 *	21.20 ± 0.73 *	3.2 ± 0.12 *	8.7 ± 0.51 ***	5.4 ± 0.10 *	0.35 ± 0.07	0.98 ± 0.02 **	2.5 ± 0.133
Strain 155	25.90 ± 0.60	19.4 ± 0.24 ***	2.7 ± 0.2	7.4 ± 0.76 **	4.8 ± 0.2	0.5 ± 0.0.37	0.66 ± 0.14	2.5 ± 0.133

Data represent means ± standard error (*n* = 8). * statistical significance Tukey *p* < 0.05, ** 0.01, and *** 0.001.

## Data Availability

Data is contained within the article or Appendix A.

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
