# Peer review of "Bacterial Strains from Saline Environment Modulate the Expression of Saline Stress-Responsive Genes in Pepper (Capsicum annuum)"

_plants, 2023, doi:10.3390/plants12203576_

Round 1
Reviewer 1 Report (Previous Reviewer 2)
Dear Author,
Please find the comments below:
1. Give the title to the experiment no. 1 for easy to to understand the work carry out in that.
2. Give the title to the experiment no. 2 for easy to to understand the work carry out in that.
3. Conclusion must be rewrite
4. few comments mentioned in attached PDF.

Author Response
Dear Reviewer,
Suggested changes and corrections were made to the manuscript. The highlighted areas were removed as editor suggested.
Dear Author,
Please find the comments below:
- Give the title to the experiment no. 1 for easy to to understand the work carry out in that.
Done. Line 133.
- Give the title to the experiment no. 2 for easy to to understand the work carry out in that.
Done. Line 151.
- Conclusion must be rewrite.
Done. Lines 350-359.
- Few comments mentioned in attached PDF.
Done. Lines 82, 94, 101-102, 120, 350-359.

Reviewer 2 Report (New Reviewer)
Comments to the Author:
Title: Bacterial strains from saline environment modulate the expression of saline stress-responsive genes in pepper (Capsicum annuum)
Overview and general recommendation:
The manuscript deals with an important topic related to the modulation of saline stress-responsive genes in pepper (Capsicum annuum) using bacterial strains from saline environment. The manuscript technically sounds well and shows high novelty. However, it needs major linguistic adjustments; therefore, I invite the authors to pass their manuscript to a native English speaker for editing and revision. In this regard, the needed adjustments are highlighted in “Minor comments” section. Also, numerous statements lack reliable sources (references) that should be provided.
The Abstract section outlines clearly the problematic, aims, methodology and findings of the current study while reporting the main conclusions aroused. However, it should enclose the percentage of observed improvements.
The Introduction section is well structured and aiming and underlines appropriately the whole subject under study. However, the aims of the study shall be better highlighted. The Materials and methods section is clear, well written, and encloses all the information related to the adopted methodology, and statistical analysis. Although it shows a correct statistical representation, the Results part needs major adjustments. The scientific analysis of the findings should be well improved. Percentages of improvements should be highlighted. On the other hand, the findings of the study were generally well discussed in the Discussion section by relying on previous published studies in literature. An appropriate Conclusions section was added in which authors summarized appropriately the findings of their study. However, they shall suggest further related research being based on the raised assumptions.
My comments and queries for authors are detailed below in “Major comments” and “Minor comments” sections.
1.1. Major comments:
1- The manuscript needs major linguistic adjustments; accordingly, I invite the authors to pass their manuscript to a native English speaker for editing and revision. Most needed adjustments are highlighted in “Minor comments” section.
2- Numerous statements lack reliable sources (references) that should be provided.
3- Abstract: The Abstract part should enclose percentage of improvements.
4- 1. Introduction: The aims of the study shall be better highlighted.
5- 3. Results: The scientific analysis of the findings shall be improved. Percentages of improvements should be highlighted.
1.2. Minor comments:
6- Abstract: Page 1, line 19: Kindly adjust the sentence as follow: “The aim of this study was to evaluate…”
7- Abstract: Page 1, line 22: Kindly mention the percentage of improvement.
8- 1. Introduction: Page 1, lines 36–39: “Stress… produced”: This statement lacks reliable sources (references); accordingly, kindly add the following suitable one: “doi:10.3390/plants12051115”.
9- 1. Introduction: Page 1, lines 39–42: “Sophisticated… damage”: The sentence is cumbersome; accordingly, kindly reformulate in order to make it clearer and more aiming.
10- 1. Introduction: Page 1, lines 39–44: “Sophisticated… manner”: These statements lack reliable sources (references); accordingly, kindly provide them.
11- 1. Introduction: Page 2, lines 52–54: “On the other… in plants”: Same recommendation as in the previous comment.
12- 1. Introduction: Page 2, lines 60–62: “An alternative… tolerance”: This statement lacks reliable sources (references); accordingly, kindly add the following suitable one: “doi:10.3390/horticulturae8090830”.
13- 1. Introduction: Page 2, lines 62–66: “PGPB… expression”: These statements lack reliable sources (references); accordingly, kindly provide them.
14- 1. Introduction: Page 2, lines 68–72: “Among… [10,11]”: The sentence is cumbersome; accordingly, kindly reformulate in order to make it clearer and more aiming.
15- 1. Introduction: Page 2, lines 73–76: “Physical… by PGPB”: This statement lacks reliable sources (references); accordingly, kindly provide them.
16- 1. Introduction: Page 2, line 83: Kindly adjust as follow: “mitigated”.
17- 1. Introduction: Page 2, lines 90–93: “In a previous… [15]”: The sentence is cumbersome; accordingly, kindly reformulate in order to make it clearer and more aiming.
18- 2. Materials and methods, 2.2. Evaluation of bacterial strains for growth promotion in Capsicum annuum: Page 3, lines 106–108: “The seeds… 30 days”: Same recommendation as in the previous comment.
19- 2. Materials and methods, 2.2. Evaluation of bacterial strains for growth promotion in Capsicum annuum: Page 3, line 109: Is it a 50% or full Hoagland solution? Kindly mention that.
20- 2. Materials and methods, 2.2. Evaluation of bacterial strains for growth promotion in Capsicum annuum: Page 3, lines 109–111: “When… inoculation”: The sentence is badly written in standard English; accordingly, kindly reformulate it.
21- 2. Materials and methods, 2.2. Evaluation of bacterial strains for growth promotion in Capsicum annuum: Page 3, lines 112 and 116: Kindly adjust as follow: “bacterial”.
22- 2. Materials and methods, 2.2. Evaluation of bacterial strains for growth promotion in Capsicum annuum: Page 3, lines 113–114: Kindly adjust as follow: “℃”.
23- 2. Materials and methods, 2.2. Evaluation of bacterial strains for growth promotion in Capsicum annuum: Page 3, line 117: Kindly adjust as follow: “Plant”.
24- 2. Materials and methods, 2.2. Evaluation of bacterial strains for growth promotion in Capsicum annuum: Page 3, lines 120–122: “Plants… experiment”: The sentence is cumbersome; accordingly, kindly reformulate in order to make it clearer and more aiming.
25- 2. Materials and methods, 2.2. Evaluation of bacterial strains for growth promotion in Capsicum annuum: Page 3, lines 127–128: “Plants… experiment”: The sentence is unclear; accordingly, kindly reformulate it in a more appropriate manner.
26- 2. Materials and methods, 2.3. Stress salinity and bacteria strains induced expression genes: Page 3, line 137: Kindly adjust as follow: “mL”.
27- 2. Materials and methods, 2.3. Stress salinity and bacteria strains induced expression genes: Page 3, lines 140–144: “Each… NaCl”: Kindly reformulate the sentence in a more appropriate manner.
28- 2. Materials and methods, 2.3. Stress salinity and bacteria strains induced expression genes: Pages 3–4, lines 144, 146–147, 153–154, and 161: Kindly adjust as follow: “bacterial”.
29- 2. Materials and methods, 2.3. Stress salinity and bacteria strains induced expression genes: Page 4, line 154: Kindly adjust as follow: “mL”.
30- 2. Materials and methods, 2.3. Stress salinity and bacteria strains induced expression genes: Page 4, line 161: Kindly adjust as follow: “initiation”.
31- 2. Materials and methods, 2.4. RNA extraction: Page 4, line 170: Kindly add put a space between the number and the temperature’ unit.
32- 2. Materials and methods, 2.5. Relative genes expression levels by qPCR analysis: Page 4, line 171: Kindly adjust as follow: “expression”.
33- 2. Materials and methods, 2.5. Relative genes expression levels by qPCR analysis: Page 4, line 177: Kindly put a space between the number and the temperature’s unit.
34- 2. Materials and methods, 2.5. Relative genes expression levels by qPCR analysis: Page 4, lines 181–182: Kindly adjust the equation’s (formula’s) presentation following the journal’s template.
35- 2. Materials and methods, 2.6. Statistical analysis: Page 4, line 188: Kindly adjust as follow: “two-way”.
36- 2. Materials and methods, 2.6. Statistical analysis: Page 4, line 189: Kindly adjust as follow: “done using”.
37- 3. Results, 3.1. Effect of bacteria isolated from saline habitats inoculation on C. annuum growth: Page 4, lines 193–195: “The bacterial… [16,19]”: The sentence is badly written in standard English; accordingly, kindly reformulate it.
38- 3. Results, 3.2. Expression analysis of stress-related genes in plants inoculated with the bacterial strains from saline sites: Page 5, lines 202–204: “For Bacillus… in expression”: The sentence is cumbersome; accordingly, kindly reformulate in order to make it clearer and more aiming.
39- 3. Results, 3.2. Expression analysis of stress-related genes in plants inoculated with the bacterial strains from saline sites: Page 5, line 204: Kindly adjust as follow: “The effect”.
40- 3. Results, 3.2. Expression analysis of stress-related genes in plants inoculated with the bacterial strains from saline sites: Page 5, line 207: Kindly adjust as follow: “showed”.
41- 3. Results, 3.2. Expression analysis of stress-related genes in plants inoculated with the bacterial strains from saline sites: Page 5, lines 214–215: “At time… inoculants”: Kindly avoid the first voice form of the sentence and adopt the impersonal form instead.
42- 3. Results, 3.2. Expression analysis of stress-related genes in plants inoculated with the bacterial strains from saline sites: Page 5, lines 221–223: The sentence is badly written in standard English; accordingly, kindly reformulate it.
43- 3. Results, 3.3. Effect of different NaCl concentrations on gene expression of C. annuum: Page 5, lines 227–228: “A greater… 100 mM”: The sentence is cumbersome; accordingly, kindly reformulate in order to make it clearer and more aiming.
44- 3. Results, 3.3. Effect of different NaCl concentrations on gene expression of C. annuum: Page 5, lines 229–232: “For the second… time 3”: Same recommendation as in the previous comment.
45- 3. Results, 3.3. Effect of different NaCl concentrations on gene expression of C. annuum: Page 5, lines 236–237: Kindly adjust as follow: “times”.
46- 3. Results, 3.3. Effect of different NaCl concentrations on gene expression of C. annuum: Page 5, line 240: Kindly adjust as follow: “indicated”.
47- 3. Results, 3.4. Expression analysis of genes associated with salt tolerance in plants with dual treatment, 3.4.1. Bacterial inoculation after salt stress: Page 5, line 243: Kindly adjust as follow: “bacterial”.
48- 3. Results, 3.4. Expression analysis of genes associated with salt tolerance in plants with dual treatment, 3.4.1. Bacterial inoculation after salt stress: Page 6, line 247: Kindly adjust as follow: “was that of the genes”.
49- 3. Results, 3.4. Expression analysis of genes associated with salt tolerance in plants with dual treatment, 3.4.1. Bacterial inoculation after salt stress: Page 6, lines 260–262: “The CaMYB72… (Figure 5g)”: The sentence is cumbersome; accordingly, kindly reformulate in order to make it clearer and more aiming.
50- 3. Results, 3.4. Expression analysis of genes associated with salt tolerance in plants with dual treatment, 3.4.2. Bacterial inoculation before salt stress: Page 6, line 262: Kindly adjust as follow: “Bacterial”.
51- 4. Discussion: Page 6, line 280: Kindly adjust as follow: “Under highly saline conditions”.
52- 4. Discussion: Page 6, lines 287–289: “In the present… [15]”: The sentence is badly written in standard English; accordingly, kindly reformulate it.
53- 4. Discussion: Page 6, line 293: Kindly adjust as follow: “The present study”.
54- 4. Discussion: Page 7, lines 298–300: “All… this work”: The sentence is cumbersome; accordingly, kindly reformulate in order to make it clearer and more aiming.
55- 4. Discussion: Page 7, line 301: Kindly adjust as follow: “registered”.
56- 4. Discussion: Page 7, lines 301–302: “Few… family”: This statement lacks reliable sources (references); accordingly, kindly provide them.
57- 4. Discussion: Page 7, lines 302–307: “An opposite… species”: These sentences are even cumbersome or badly written in standard English; accordingly, kindly reformulate in order to make them clearer and more aiming.
58- 4. Discussion: Page 7, line 312: Kindly replace the comma “,” by a semi-column “;”.
59- 4. Discussion: Page 7, lines 315–316: “In this… responses”: Kindly avoid the first voice form of the sentence and adopt the impersonal form instead.
60- 4. Discussion: Page 7, line 319: Kindly adjust as follow: “under current experimental”.
61- 4. Discussion: Page 7, lines 321–326: “Nevertheless… [9,40,41]”: The sentence is long and cumbersome; accordingly, kindly reformulate in order to make it more concise, clearer and more aiming.
62- 4. Discussion: Page 7, lines 326–328: “On the other… downregulation”: The sentence is badly written in standard English; accordingly, kindly reformulate it.
63- 4. Discussion: Page 7, line 330: Kindly adjust as follow: “conferred”.
64- 4. Discussion: Page 7, lines 336–337: “The ROS… conditions”: This statement lacks reliable sources (references); accordingly, kindly provide them.
65- 4. Discussion: Page 7, line 339: Kindly adjust as follow: “hrs”.
66- 4. Discussion: Page 7, lines 339–340: Kindly adjust as follow: “in this study”.
67- 4. Discussion: Page 7, line 341: Kindly adjust as follow: “and this gene”.
68- 4. Discussion: Page 7, line 348: Kindly adjust as follow: “under saline”.
69- 5. Conclusions: Page 8, lines 353–357: “In the present… C. annuum”: Kindly adjust as follow: “evaluated”. Moreover, kindly avoid the first voice form of the sentence and adopt the impersonal form instead.
70- 5. Conclusions: Page 8, line 360: Kindly adjust as follow: “indicated”.
71- 5. Conclusions: Page 8, line 362: Kindly add a sentence here in which you suggest further related research being based on the raised assumptions from the current study.
The manuscript needs major linguistic adjustments; accordingly, I kindly ask the authors to pass their manuscript to a native English speaker for editing and revision. Most needed adjustments are highlighted in "Minor comments" section of my enclosed report.
Author Response
Dear Reviewer,
Suggested changes and corrections were made to the manuscript. The highlighted areas were removed as editor suggested.
1.1. Major comments:
1- The manuscript needs major linguistic adjustments; accordingly, I invite the authors to pass their manuscript to a native English speaker for editing and revision. Most needed adjustments are highlighted in “Minor comments” section.
Done.
2- Numerous statements lack reliable sources (references) that should be provided.
Done.
3- Abstract: The Abstract part should enclose percentage of improvements.
Done. Lines 22-23.
4- 1. Introduction: The aims of the study shall be better highlighted.
Done. Lines 95-97.
5- 3. Results: The scientific analysis of the findings shall be improved. Percentages of improvements should be highlighted.
Done. Lines 195-199.
1.2. Minor comments:
6- Abstract: Page 1, line 19: Kindly adjust the sentence as follow: “The aim of this study was to evaluate…” Done. Line 19.
7- Abstract: Page 1, line 22: Kindly mention the percentage of improvement. Done. Lines 22-23.
8- 1. Introduction: Page 1, lines 36–39: “Stress… produced”: This statement lacks reliable sources (references); accordingly, kindly add the following suitable one: “doi:10.3390/plants12051115”. Done. Line 39.
9- 1. Introduction: Page 1, lines 39–42: “Sophisticated… damage”: The sentence is cumbersome; accordingly, kindly reformulate in order to make it clearer and more aiming. Done. Lines 39-42.
10- 1. Introduction: Page 1, lines 39–44: “Sophisticated… manner”: These statements lack reliable sources (references); accordingly, kindly provide them. Done. Line 42.
11- 1. Introduction: Page 2, lines 52–54: “On the other… in plants”: Same recommendation as in the previous comment. Done. Line 54.
12- 1. Introduction: Page 2, lines 60–62: “An alternative… tolerance”: This statement lacks reliable sources (references); accordingly, kindly add the following suitable one: “doi:10.3390/horticulturae8090830”. Done. Line 62.
13- 1. Introduction: Page 2, lines 62–66: “PGPB… expression”: These statements lack reliable sources (references); accordingly, kindly provide them. Done. Lines 66, 68.
14- 1. Introduction: Page 2, lines 68–72: “Among… [10,11]”: The sentence is cumbersome; accordingly, kindly reformulate in order to make it clearer and more aiming. Done. Lines 68-72.
15- 1. Introduction: Page 2, lines 73–76: “Physical… by PGPB”: This statement lacks reliable sources (references); accordingly, kindly provide them. Done. Line 76.
16- 1. Introduction: Page 2, line 83: Kindly adjust as follow: “mitigated”. Done. Line 83.
17- 1. Introduction: Page 2, lines 90–93: “In a previous… [15]”: The sentence is cumbersome; accordingly, kindly reformulate in order to make it clearer and more aiming. Done. Lines 91-94.
18- 2. Materials and methods, 2.2. Evaluation of bacterial strains for growth promotion in Capsicum annuum: Page 3, lines 106–108: “The seeds… 30 days”: Same recommendation as in the previous comment. Done. Lines 106-108.
19- 2. Materials and methods, 2.2. Evaluation of bacterial strains for growth promotion in Capsicum annuum: Page 3, line 109: Is it a 50% or full Hoagland solution? Kindly mention that. Done. Line 109.
20- 2. Materials and methods, 2.2. Evaluation of bacterial strains for growth promotion in Capsicum annuum: Page 3, lines 109–111: “When… inoculation”: The sentence is badly written in standard English; accordingly, kindly reformulate it. Done. Lines 108-111.
21- 2. Materials and methods, 2.2. Evaluation of bacterial strains for growth promotion in Capsicum annuum: Page 3, lines 112 and 116: Kindly adjust as follow: “bacterial”. Done. Lines 112, 116.
22- 2. Materials and methods, 2.2. Evaluation of bacterial strains for growth promotion in Capsicum annuum: Page 3, lines 113–114: Kindly adjust as follow: “℃”. Done. Lines 108, 113, 114, 129.
23- 2. Materials and methods, 2.2. Evaluation of bacterial strains for growth promotion in Capsicum annuum: Page 3, line 117: Kindly adjust as follow: “Plant”. Done. Line 117.
24- 2. Materials and methods, 2.2. Evaluation of bacterial strains for growth promotion in Capsicum annuum: Page 3, lines 120–122: “Plants… experiment”: The sentence is cumbersome; accordingly, kindly reformulate in order to make it clearer and more aiming. Done. Lines 120-122.
25- 2. Materials and methods, 2.2. Evaluation of bacterial strains for growth promotion in Capsicum annuum: Page 3, lines 127–128: “Plants… experiment”: The sentence is unclear; accordingly, kindly reformulate it in a more appropriate manner. Done. Lines 127-130.
26- 2. Materials and methods, 2.3. Stress salinity and bacteria strains induced expression genes: Page 3, line 137: Kindly adjust as follow: “mL”. Done. Line 136.
27- 2. Materials and methods, 2.3. Stress salinity and bacteria strains induced expression genes: Page 3, lines 140–144: “Each… NaCl”: Kindly reformulate the sentence in a more appropriate manner. Done. Lines 139-142.
28- 2. Materials and methods, 2.3. Stress salinity and bacteria strains induced expression genes: Pages 3–4, lines 144, 146–147, 153–154, and 161: Kindly adjust as follow: “bacterial”. Done. Lines 145, 146, 147, 151, 153.
29- 2. Materials and methods, 2.3. Stress salinity and bacteria strains induced expression genes: Page 4, line 154: Kindly adjust as follow: “mL”. Done. Line 153.
30- 2. Materials and methods, 2.3. Stress salinity and bacteria strains induced expression genes: Page 4, line 161: Kindly adjust as follow: “initiation”. Done. Line 160.
31- 2. Materials and methods, 2.4. RNA extraction: Page 4, line 170: Kindly add put a space between the number and the temperature’ unit. Done. Line 169.
32- 2. Materials and methods, 2.5. Relative genes expression levels by qPCR analysis: Page 4, line 171: Kindly adjust as follow: “expression”. Done. Line 170.
33- 2. Materials and methods, 2.5. Relative genes expression levels by qPCR analysis: Page 4, line 177: Kindly put a space between the number and the temperature’s unit. Done. Lines 175-176.
34- 2. Materials and methods, 2.5. Relative genes expression levels by qPCR analysis: Page 4, lines 181–182: Kindly adjust the equation’s (formula’s) presentation following the journal’s template. The equation is in a usually used format in gene expression analysis. Lines 179-181.
35- 2. Materials and methods, 2.6. Statistical analysis: Page 4, line 188: Kindly adjust as follow: “two-way”. Done. Line 187.
36- 2. Materials and methods, 2.6. Statistical analysis: Page 4, line 189: Kindly adjust as follow: “done using”. Done. Line 188.
37- 3. Results, 3.1. Effect of bacteria isolated from saline habitats inoculation on C. annuum growth: Page 4, lines 193–195: “The bacterial… [16,19]”: The sentence is badly written in standard English; accordingly, kindly reformulate it. Done. Lines 192-193.
38- 3. Results, 3.2. Expression analysis of stress-related genes in plants inoculated with the bacterial strains from saline sites: Page 5, lines 202–204: “For Bacillus… in expression”: The sentence is cumbersome; accordingly, kindly reformulate in order to make it clearer and more aiming. Done. Lines 201-203.
39- 3. Results, 3.2. Expression analysis of stress-related genes in plants inoculated with the bacterial strains from saline sites: Page 5, line 204: Kindly adjust as follow: “The effect”. Done. Line 203.
40- 3. Results, 3.2. Expression analysis of stress-related genes in plants inoculated with the bacterial strains from saline sites: Page 5, line 207: Kindly adjust as follow: “showed”. Done. Line 206.
41- 3. Results, 3.2. Expression analysis of stress-related genes in plants inoculated with the bacterial strains from saline sites: Page 5, lines 214–215: “At time… inoculants”: Kindly avoid the first voice form of the sentence and adopt the impersonal form instead. Done. Lines 213-214.
42- 3. Results, 3.2. Expression analysis of stress-related genes in plants inoculated with the bacterial strains from saline sites: Page 5, lines 221–223: The sentence is badly written in standard English; accordingly, kindly reformulate it. Done. Lines 219-222.
43- 3. Results, 3.3. Effect of different NaCl concentrations on gene expression of C. annuum: Page 5, lines 227–228: “A greater… 100 mM”: The sentence is cumbersome; accordingly, kindly reformulate in order to make it clearer and more aiming. Done. Lines 226-227.
44- 3. Results, 3.3. Effect of different NaCl concentrations on gene expression of C. annuum: Page 5, lines 229–232: “For the second… time 3”: Same recommendation as in the previous comment. Done. Lines 229-231.
45- 3. Results, 3.3. Effect of different NaCl concentrations on gene expression of C. annuum: Page 5, lines 236–237: Kindly adjust as follow: “times”. Done. Line 236.
46- 3. Results, 3.3. Effect of different NaCl concentrations on gene expression of C. annuum: Page 5, line 240: Kindly adjust as follow: “indicated”. Done. Line 239.
47- 3. Results, 3.4. Expression analysis of genes associated with salt tolerance in plants with dual treatment, 3.4.1. Bacterial inoculation after salt stress: Page 5, line 243: Kindly adjust as follow: “bacterial”. Done. Line 242.
48- 3. Results, 3.4. Expression analysis of genes associated with salt tolerance in plants with dual treatment, 3.4.1. Bacterial inoculation after salt stress: Page 6, line 247: Kindly adjust as follow: “was that of the genes”. Done. Line 246.
49- 3. Results, 3.4. Expression analysis of genes associated with salt tolerance in plants with dual treatment, 3.4.1. Bacterial inoculation after salt stress: Page 6, lines 260–262: “The CaMYB72… (Figure 5g)”: The sentence is cumbersome; accordingly, kindly reformulate in order to make it clearer and more aiming. Done. Lines 259-261.
50- 3. Results, 3.4. Expression analysis of genes associated with salt tolerance in plants with dual treatment, 3.4.2. Bacterial inoculation before salt stress: Page 6, line 262: Kindly adjust as follow: “Bacterial”. Done. Line 267.
51- 4. Discussion: Page 6, line 280: Kindly adjust as follow: “Under highly saline conditions”. Done. Line 279.
52- 4. Discussion: Page 6, lines 287–289: “In the present… [15]”: The sentence is badly written in standard English; accordingly, kindly reformulate it. Done. Done. Lines 286-288.
53- 4. Discussion: Page 6, line 293: Kindly adjust as follow: “The present study”. Done. Line 292.
54- 4. Discussion: Page 7, lines 298–300: “All… this work”: The sentence is cumbersome; accordingly, kindly reformulate in order to make it clearer and more aiming. Done. Lines 297-301.
55- 4. Discussion: Page 7, line 301: Kindly adjust as follow: “registered”. Done. Line 300.
56- 4. Discussion: Page 7, lines 301–302: “Few… family”: This statement lacks reliable sources (references); accordingly, kindly provide them. Done. Line 301.
57- 4. Discussion: Page 7, lines 302–307: “An opposite… species”: These sentences are even cumbersome or badly written in standard English; accordingly, kindly reformulate in order to make them clearer and more aiming. Done. Lines 301-304.
58- 4. Discussion: Page 7, line 312: Kindly replace the comma “,” by a semi-column “;”. Done. Line 309.
59- 4. Discussion: Page 7, lines 315–316: “In this… responses”: Kindly avoid the first voice form of the sentence and adopt the impersonal form instead. Done. Lines 312-314.
60- 4. Discussion: Page 7, line 319: Kindly adjust as follow: “under current experimental”. Done. Line 317.
61- 4. Discussion: Page 7, lines 321–326: “Nevertheless… [9,40,41]”: The sentence is long and cumbersome; accordingly, kindly reformulate in order to make it more concise, clearer and more aiming. Done. Lines 318-324.
62- 4. Discussion: Page 7, lines 326–328: “On the other… downregulation”: The sentence is badly written in standard English; accordingly, kindly reformulate it. Done. Lines 324-328.
63- 4. Discussion: Page 7, line 330: Kindly adjust as follow: “conferred”. Done. 327.
64- 4. Discussion: Page 7, lines 336–337: “The ROS… conditions”: This statement lacks reliable sources (references); accordingly, kindly provide them. Done. Line 334.
65- 4. Discussion: Page 7, line 339: Kindly adjust as follow: “hrs”. Done. Line 336.
66- 4. Discussion: Page 7, lines 339–340: Kindly adjust as follow: “in this study”. Done. Line 336.
67- 4. Discussion: Page 7, line 341: Kindly adjust as follow: “and this gene”. Done. Line 338.
68- 4. Discussion: Page 7, line 348: Kindly adjust as follow: “under saline”. Done. Line 344.
69- 5. Conclusions: Page 8, lines 353–357: “In the present… C. annuum”: Kindly adjust as follow: “evaluated”. Moreover, kindly avoid the first voice form of the sentence and adopt the impersonal form instead. Done. Lines 350-352.
70- 5. Conclusions: Page 8, line 360: Kindly adjust as follow: “indicated”. Done. Line 355.
71- 5. Conclusions: Page 8, line 362: Kindly add a sentence here in which you suggest further related research being based on the raised assumptions from the current study. Done. Lines 356-357.
Comments on the Quality of English Language
The manuscript needs major linguistic adjustments; accordingly, I kindly ask the authors to pass their manuscript to a native English speaker for editing and revision. Most needed adjustments are highlighted in "Minor comments" section of my enclosed report. Done.

Round 2
Reviewer 2 Report (New Reviewer)
Comments to the Author:
Title: Bacterial strains from saline environment modulate the expression of saline stress-responsive genes in pepper (Capsicum annuum)
Overview and general recommendation:
Authors have made significant improvement to their manuscript and are well thanked for that. Although some minor typo mistakes are detected, they can be adjusted during pre-proof stage. Therefore, based on the overall evaluation of the manuscript, I find it well suitable now for publication in current form.
Although some minor typo mistakes are detected, they can be adjusted during pre-proof stage.
This manuscript is a resubmission of an earlier submission. The following is a list of the peer review reports and author responses from that submission.
Round 1
Reviewer 1 Report
The manuscript provides valuable information on the modulation of several genes responsive to saline stress when two bacteria from a saline environment are applied. The growth-promoting effect of pepper plants due to microbial treatment is evident. The authors presented the expression profile of various stress-related genes in response to microbes, salt, and a combination of microbes and salt treatments in pepper plants. The overall experimental design, involving the treatment of plants with bacteria and salts at 15-day intervals (labeled as T1, T2, etc.), is intriguing. However, several significant flaws should be addressed through logical reasoning and the inclusion of additional experimental results.
1. Authors claim in the abstract (Line 16) that the microbial application can confer stress tolerance in plants under saline soil conditions. However, to support this claim, it is essential to demonstrate improved growth or growth rescue of the plants when exposed to salt stress conditions and treated with the microbes. Therefore, it is crucial to include relevant experiments or data that assess the actual effect of the bacterial application on the salt tolerance of pepper plants. This step is necessary to ensure the significance and validity of the subsequent gene expression study.
2. The rationale for choosing the three bacteria used in this study is missing. The selection of the bacteria appears to be arbitrary, lacking relevant information regarding their abundance in the rhizosphere of salt-tolerant plant varieties or any bioinformatic analyses regarding their genomic potential for providing salt-tolerant services. In this study, the authors randomly sourced two bacteria from high-salinity ponds, which do not have any direct connection to the plant association.
3. The author solely focused on the growth benefits of pepper plants resulting from the application of the selected bacteria. However, the influence of the natural soil microbiome was not considered due to the prior autoclaving of the soil before use. To validate the effects of the selected bacteria in natural conditions, it is necessary to test the growth benefits and salt tolerance (if any) of the plants in natural soils with the existing microbiome.
4. Most genes have pleiotropic effects, so the up-or down-regulation of one or two genes will not necessarily indicate similar expression of the entire pathway. On the other hand, RNAseq analysis can provide a comprehensive view of global transcriptional regulation in a specific situation.
5. The overall representation of the data in the figures was not up to the mark, making it difficult to view and analyze.
6. In Results (Line 230) ‘expression genes in general in time 1 showed a null or negative response to inoculation’: Several genes like CaLOX2, CaMYB72, CaWRKY, CaDNaJ02, in fig1 & 2 showed upregulation at T1. Among all of them, why did the authors only pointed out the upregulation of CaHsfA2 in the result (Line 233)?
Overall the quality of English is good, but moderate edits may require.
Reviewer 2 Report
Please see the attached PDF with comments
